# The Biology of Classic Hairy Cell Leukemia

**DOI:** 10.3390/ijms22157780

**Published:** 2021-07-21

**Authors:** Jan-Paul Bohn, Stefan Salcher, Andreas Pircher, Gerold Untergasser, Dominik Wolf

**Affiliations:** 1Department of Internal Medicine V, Hematology and Oncology, Medical University of Innsbruck, 6020 Innsbruck, Austria; stefan.salcher@i-med.ac.at (S.S.); andreas.pircher@i-med.ac.at (A.P.); gerold.untergasser@i-med.ac.at (G.U.); dominik.wolf@i-med.ac.at (D.W.); 2Experimental Oncogenomic Group, Tyrolean Cancer Research Institute, 6020 Innsbruck, Austria; 3Department of Hematology and Oncology, Medical Clinic 3, University Hospital Bonn, 53127 Bonn, Germany

**Keywords:** hairy cell leukemia, HCL, biology, microenvironment, BRAF V600E, DUSP, single-cell sequencing, vitronectin, fibronectin, JNK, p38, B-cell receptor, epigenetic, methylome, microRNA

## Abstract

Classic hairy cell leukemia (HCL) is a rare mature B-cell malignancy associated with pancytopenia and infectious complications due to progressive infiltration of the bone marrow and spleen. Despite tremendous therapeutic advances achieved with the implementation of purine analogues such as cladribine into clinical practice, the culprit biologic alterations driving this fascinating hematologic disease have long stayed concealed. Nearly 10 years ago, BRAF V600E was finally identified as a key activating mutation detectable in almost all HCL patients and throughout the entire course of the disease. However, additional oncogenic biologic features seem mandatory to enable HCL transformation, an open issue still under active investigation. This review summarizes the current understanding of key pathogenic mechanisms implicated in HCL and discusses major hurdles to overcome in the context of other BRAF-mutated malignancies.

## 1. Introduction

Classic hairy cell leukemia (HCL) is an uncommon chronic lymphoproliferative disorder characterized by progressive bone marrow failure due to infiltrating malignant B cells with “hairy-like surface projections” provoking frequent infectious complications [1]. Although associated with a dismal survival prognosis when first described in 1958 [2], the therapeutic implementation of purine analogues such as cladribine as early as in the 1990s has turned HCL into one of the biggest treatment successes in cancer history [3]. Nowadays most HCL patients may face a near normal life span when compared to the general population [4]. Despite these tremendous therapeutic advances in the last century, however, the biology underlying this fascinating hematologic disease has long remained obscure. Before receiving the morphologically descriptive title of HCL by Schreck and Donelly in 1966 [5], the disease was originally termed *Leukemic reticuloendotheliosis*, reflecting its distinct pattern of tissue infiltration: HCL cells typically accumulate in the bone marrow, the red pulp of the spleen, the hepatic sinusoids and portal tracts, but usually spare significant lymph node involvement [2]. Thereby, both terminologies each highlight two major forces driving this unique disease phenotype: the malignant B cell itself and nourishing extracellular stimuli provided by a resource-rich local tissue microenvironment. Novel insights have shed light onto the cellular origin, the transforming events as well as the HCL cells’ beneficious crosstalk with the local tissue microenvironment. This review aims to summarize our current understanding of the major biologic footprints in HCL and discusses relevant obstacles to overcome in the context of other BRAF-mutated malignancies.

## 2. Mature Clonal B Cells with ‘Hairy’ Cytoplasmatic Projections

The initial classification as *leukemic reticuloendotheliosis* illustrates the original assumption that HCL cells were of myeloid ancestry due to similarities in cell morphology and function reminiscent to those of the mononuclear macrophage system [6]. HCL cells share the unusual feature for lymphocytes of a certain phagocytic activity when exposed to bacteria [7]. Contrary to normal B-cells and other lymphomas, HCL cells overexpress various cytoskeleton components such as actins, intracellular phosphoproteins as well as members of the Rho family of small GPTases involved in active cytoskeletal re-organization considered the basis for the cells’ dynamic ‘hairy’ morphology and observed phagocytic activity [8,9]. Immunophenotyping studies, however, clearly demonstrated that HCL cells are monoclonal B cells arrested at a late stage of differentiation showing a strong light-chain restricted surface immunoglobulin along with a typical immunophenotype (i.e., CD103+, CD25+, CD11c+, CD123+) [10,11]. Unique for B cell neoplasms, HCL cells co-express dual, clonally related Ig-isotypes (IgM, IgG, and IgA) in up to 80% of patients [12]. Neither morphologically nor immunophenotypically HCL cells are reminiscent of any healthy B cell subpopulation, and therefore their normal counterpart has long been unclear. However, frequent expression of rearranged immunoglobulin variable (IGHV) genes with somatic mutations, a genome-wide expression profile closely related to post germinal center memory B cells and ultimately recently performed methylome analysis, strongly indicate that the cell of origin must have transited through the germinal center [13,14,15]. Moreover, comparative sequence hybridization studies linked the malignant cells to a splenic expression signature on a genome-wide basis indicating that the normal counterpart of HCL may be located in the spleen [16].

Although infiltration of the spleen is a characteristic finding in HCL, splenomegaly is a result of red-pulp hypertrophy, whereas the white pulp containing the suspected counterpart memory B cells even becomes atrophic [2], still questioning this hypothesis. Ultimately, the neoplastic transforming events *per se* may account for the challenges in identifying the cell of origin.

## 3. Malignant Cell Inherent Features Driving Classic Hairy Cell Leukemia

Consistent with the postulated post germinal center origin, HCL cells lack recurrent chromosomal aberrations and the genetic events driving malignant cell transformation have long been unclear. In contrast to most other cancer types, gene expression analysis of HCL revealed a rather strict conservation of pathways involved in proliferation and survival when compared to memory B cells. HCL disease progression is mainly facilitated by extended cell survival rather than increased proliferative cell fractions [14]. 

Finally, in 2011, Tiacci et al. [17] identified a gain-of-function mutation of the B-rapidly accelerated fibrosarcoma (BRAF) serine/threonine protein kinase (V600E) detectably in almost all HCL patients. The activating BRAF mutation is a central genetic driver in HCL cells, as it is detected in the entire tumor clone and is highly stable at relapse. Moreover, when inhibited, HCL lose their typical gene expression signature, their hairy morphology and, ultimately, the cells die when exposed to BRAF- and mitogen-activated protein kinase kinases (MEK)-inhibitors [18]. Meanwhile, several clinical trials have demonstrated encouraging efficacy of BRAF- and MEK-inhibitors in relapsed and/or refractory HCL patients [19,20]. Superior in terms of drug tolerability when compared with chemo(immuno)therapy, BRAF- and MEK-inhibitors already serve as a valuable therapeutic alternative for HCL patients not qualifying for standard chemo(immuno)therapy, i.e., due to active infections, comorbidities and/or advanced age [21].

Remarkably, the BRAF gain-of-function mutation occurs already in earlier differentiation stages, including hematopoietic stem cells (HSC) or B-cell lymphoid progenitors of affected patients. When BRAF-mutated HSC were transplanted into immunodeficient mice, however, none of those actually developed the full phenotypic picture of HCL suggesting the requirement of additional cooperating genetic alterations [22]. In a larger cohort of HCL patients (*n* = 53), copy number loss of chromosome 7q was suggested as the most frequently detected genetic lesion besides BRAF V600E [23], although evidence seems conflicting with a reported detection rate of only 8% (5/63) in another study [24]. The deleted chromosomal region involves BRAF (7q43) leading to loss of heterozygosity of the mutant BRAF allele as illustrated by a higher BRAF V600E variant allele frequency [23]. Until now, however, the potential biological implications of hemizygous BRAF-mutated versus heterozygous BRAF-mutated HCL remain incompletely understood.

Provocatively, more than 80% of benign naevi of the skin already carry oncogenic BRAF mutations while never experiencing malignant transformation [25]. Quite the opposite, reinforced extracellular-signal regulated kinases (ERK) signaling alone may rather induce senescence via upregulation of cell-cycle inhibitors such as CDKN1A (p21) and CDKN1B (p27) in human melanocytes and congenital naevi [26]. Interestingly, whole-exome sequencing studies of mostly untreated HCL patients identified an inactivating mutation of the cell cycle inhibitor CDKN1B as the second most common genetic alteration in HCL occurring in 16% (13/81) of cases [24]. Other recurrent mutations included KLF2 and KMT2C in 15% (3/20) and 15% (8/53) of patients examined, respectively [23,27]. KLF2 is a transcription factor engaged in differentiation control and B-cell homing to lymph nodes [28], possibly contributing to the distinct pattern of extra-nodal tissue distribution of HCL cells. Including KMT2C, a lysine specific histone methyltransferase, individually detected mutations seem to accumulate in genes involved in the chromatin remodeling complex such as ARID1A/ARID1B, EZH2 and KDM6A, thus, pointing towards a role for a permissive epigenetic landscape in promoting HCL biology [24] (as reviewed in Section 4).

TP53 mutations were noticed only infrequently in HCL in recent sequencing studies (0–2%) [23,29] and are typically associated with the variant form of the disease in up to 30% of cases [29,30]. Albeit, TP53 status assessment is recommended in HCL patients relapsed/refractory to standard treatment with purine analogues [31].

Taken together, these results suggest that cooperating genetic and/or epigenetic events are required to facilitate malignant transformation in BRAF V600Emutated HCL cells. Intriguingly, BRAF-mutant cell lines from melanoma and colorectal carcinoma were associated with a distinct transcriptional output including disabled feedback inhibition of the BRAF-MEK-ERK signaling pathway [32]. After exposition with the BRAF-inhibitor vemurafenib gene enrichment analysis demonstrated a sharp silencing of transcription factors associated with ERK-dependent transformation, such as members of the ETS family, FOS and MYC as well as of genes involved in MEK/ERK feedback inhibition such as dual specificity phosphatase 6 [32]. The need for further ERK signaling modulation in BRAF-driven tumors indicates that a certain balance between ERK activation and inhibition is more important than the absolute ERK signaling strength for malignant transformation. In line with this idea, dysregulated expression patterns of various ERK feedback inhibitors have been described in several cancers [33]. As such, modulated ERK signaling suppressors may act in concert to help escape ERK induced senescence and facilitate neoplastic transformation in HCL. Herein, members of the dual-specificity mitogen activated protein kinase phosphatases (DUSPs) have been strongly implicated in the regulation of oncogenic ERK signaling. DUSPs are transcriptionally regulated as downstream targets of mitogen-activated protein kinase (MAPK) signaling shown to either act as classical negative feedback regulators by dephosphorylation and inactivation of MAPKs or able to mediate cross talk between distinct MAPK pathways [34]. For instance, the inducible nuclear phosphatase DUSP5 specifically binds and inactivates ERK and is often up-regulated in tumors with activated Ras/MAPK signaling. Loss of DUSP5 was associated with inferior clinical outcomes in patients with gastric and prostate cancer and re-expression of DUSP5 shrunk cell proliferation indices in various cancer cell lines [35]. Of interest, DUSP5 knock-out mice treated with the DMBA/TPA-inducible skin carcinogenesis protocol in order to acquire Hras mutations driving skin papilloma development experienced twice as many skin cancers as DUSP5 wild-type mice [36]. In explicitly BRAF-mutant tumors, however, currently available data suggests a rather oncogenic role for DUSP5 by paradoxically increasing cytoplasmatic ERK activity, whereas DUSP5 deletion leads to BRAFV 600E-induced ERK hyperactivation and cellular senescence [37].

Cytoplasmatic DUSP6 acts as another classical negative regulator of ERK activity with an absolute substrate specificity being consistently overexpressed in response to upregulated ERK signaling in BRAF-mutant melanoma [38]. Current evidence suggests a lineage specific effect and points towards a role as an oncogenic mediator in B-cells. Inhibition of DUSP6 was associated with elevated ERK signaling resulting in an increased rate of p53-mediated apoptosis [39]. Recent gene expression profiling studies of 26 HCL patients confirmed DUSP6 as a transcriptional target of the constitutively activated BRAF-MEK-ERK pathway [18].Interestingly, DUSP6 is also regulated upon extracellular growth factor stimulation and serves as an essential mediator of fibroblast growth factor (FGF) dependent ERK activation in prenatal development [40]. FGF has been strongly implicated in HCL growth as HCL cells not only produce basic FGF, but also express its receptor FGFR1 in an autocrine manner [14]. FGFR1 activation triggers ERK signaling and drives cell-cycle progression and cell transformation in rodent fibroblasts [41].

On the contrary, the inducible nuclear phosphatase DUSP1 can bind and dephosphorylate all three major classes of MAPK (ERK, p38 and c-Jun N-terminal kinases, JNK) with a certain preference for p38 and JNK in an inhibitory fashion [42]. JNK and p38 signaling has been shown to exert a tumor suppressive function via several different ways, including p53 induced apoptosis and negative regulation of cell cycle progression [43]. Interestingly, DUSP1 expression is up-regulated particularly in low-grade tumors, where it may help escape JNK-induced apoptosis and its cellular expression levels correlate with resistance of cancer cells to chemotherapy [44]. In HCL cells, pro-apoptotic JNK and p38 signaling needs to be constantly abrogated to circumvent cell death, mainly through phosphorylation by protein kinase C (PKC) and further upstream regulated by extracellular stress signaling including cytokines and G-protein coupled receptors (Figure 1) [43]. On the contrary, reinforced p38 and JNK signaling by inhibition of PKC frequently leads to cell death in vitro. Thereby, phosphorylation of p38 and JNK is only sustained when HCL cells are cultured on vitronectin but not on a nonadherent surface, supporting a critical role of extracellular stimuli for p38 and JNK regulation in HCL biology (as reviewed in Section 5) [45]. 

Meanwhile, drug investigations of DUSP-inhibitors in cancers dependent on elevated MAPK signaling have yet not found entrance to the clinical setting. However, growing preclinical data suggests promising efficacy [46].

## 4. The Epigenetic Landscape of Classic Hairy Cell Leukemia

Considering the highly stable genomic profile in BRAF-mutated HCL, but frequently detected minimal residual disease despite treatment with BRAF inhibitors, epigenetic factors have been strongly suggested to contribute to disease biology and clinical behavior [19,47]. Epigenetic aberrations facilitate changes to the chromatin and DNA structure without altering the genetic code [48]. Thereby, DNA methylation of the cytosine residue within CpG dinucleotides is the most widely studied epigenetic modification and precludes transcriptional processes and gene expression by generating a condensed genomic structure. On the contrary, lack of DNA methylation marks favors gene activation [48]. 

Hence, DNA methylation serves as a crucial regulator of cellular differentiation and cell type specificity during hematopoietic development [49]. In healthy cells, chromosomal repetitive regions show the highest density of methylated CpG pairs, whereas isolated CpG islands commonly found in gene promotors remain unmethylated. Tumorigenesis is typically characterized by the opposite scenario with general hypomethylation and only local hypermethylation within CpG island promoters [50]. Thus, DNA methylation is increasingly recognized as a vital mechanism of neoplastic gene silencing. Deeper insights into the DNA methylation profile of HCL were first reported by Arribas and co-workers in 2019 [15]. Besides 11 BRAF-mutated HCL samples from untreated patients, the group analyzed another seven splenic marginal zone lymphoma and 21 chronic lymphocytic leukemia (CLL) samples to better characterize the overlaps and discrepancies among the mature B-cell malignancies and help identify a specific HCL methylation profile. DNA methylation data on healthy B-cell subpopulations were obtained from public databases in order to help define the normal HCL counterpart with the closest methylation profile. Genome-wide promoter methylation analysis demonstrated a distinct clustering of HCL from naïve and germinal center B-cells and revealed closest overlap with post germinal center subsets, including splenic marginal zone and memory B-cell subsets [15]. After integrating HCL gene expression profiling data derived from the pivotal study by Basso et al. [14] in 2004, gene enrichment analysis confirmed an inverse correlation between gene expression and methylation proposing DNA promotor methylation as a strong regulator of the specific HCL gene expression signature. When gene expression profiling data of memory B cells was added, integrated analysis revealed hypomethylation/overexpression of several genes associated with BCR-TLR-NF-κB, BRAF-MAPK signaling pathways as well as cell adhesion [15]. Overall, the noted methylation modifications appeared to accompany the constitutively activated BRAF-MEK-ERK pathway in HCL and proposed a permissive epigenetic machinery fueling upregulation of BRAF associated genes [15].

Of special interest, aberrant DNA methylation was also found critically involved in deregulation of microRNA (miRNA) expression [51]. In normal B cells, expression of miRNAs is strictly controlled in a stage or transformation specific fashion proposing a coordinating role in cell differentiation [52]. Recently, however, deregulated miRNA expression was identified as another crucial mechanism involved in the complex epigenetic shaping of malignant cells. The non-coding miRNAs exert their function by downregulation of target proteins and may, thus, act either in an oncogenic or tumor suppressive way [53]. In CLL, the most frequent leukemia in adults, a common example is upregulation of anti-apoptotic BCL2 due to loss of miR15a and miR16-1 in B cells with deletion 13q (del13q), the chromosomal harboring spot of these two miRNAs. Intriguingly, del13q has also been described in a minor fraction of HCL patients [23].

The role of deregulated miRNAs in HCL was first addressed by Kitagawa et al. [54] who analyzed eight patients with BRAF-mutant HCL, five patients with BRAF-wildtype splenic lymphoma with villous lymphocytes and nine patients with CLL. MiRNA expression profiling demonstrated a distinct HCL phenotype compared to the other B-cell lymphomas and identified six significantly overexpressed miRNAs, including miR-221/miR-222 family, miR-22, miR-24, miR-27a and let-7b [54]. Strikingly, miR-221/miR-222 were identified to directly target the tumor suppressor CDKN1B (p27) [55] known to counteract oncogenic BRAF-signaling in BRAF-mutated cells and drive cells towards senescence and apoptosis as is discussed above. Hence, overexpression of miR-221/miR-222 may at least in part account for the frequently found low levels of CDKN1B expression in HCL cases that do not harbor a CDKN1B loss-of-function mutation [56]. MiR-24 and miR-22 target the tumor suppressors CDKN2A (p16) and CDKN1A (p21), respectively, and may, thus, also contribute to resistance to apoptosis [57,58]. Remarkably, target prediction algorithms and subsequent pathway-enrichment analysis of overexpressed miRNAs identified the MAPK pathways as the most significantly enhanced signaling cascade and, thus, promote deregulated miRNA expression as a critical modulator of MAPK signaling in HCL [54]. Thereby, most of the predicted targets involved in MAPK signaling were part of JNK/p38 activation, whereas none was predicted for proteins of the BRAF-MEK-ERK pathway. As described above, activation of the p38-JNK pathway was demonstrated to make HCL cells susceptible to apoptosis underlining the vital role of balanced ERK and JNK/p38 signaling for HCL proliferation and survival [45]. Hence, downregulation of JNK/p38 signaling as a result of miRNA overexpression may represent another important mechanism helping HCL cells evade apoptosis [54].

## 5. Classic Hairy Cell Leukemia Depends on Extracellular Stimuli

Extracellular stimuli are provided via cellular and molecular interactions with adjacent cells, matrix proteins and cytokines in the bone marrow and spleen, altogether commonly depicted as the tissue microenvironment [59]. Nourishing crosstalk with specialized local stroma cells is vital for aggregation and persistence of HCL cells and responsible for the disease’s distinctive pattern of tissue infiltration [60]. In the bone marrow, the cellular components of the microenvironment can basically be separated into two specialized compartments: an endosteal niche where the HSCs hide in a quiescent state and the sinusoidal–vascular niche where the HSCs travel when stocking up the peripheral blood cell counts. Mesenchymal bone marrow stromal cells (BMSC), part of the endosteal compartment, persistently secrete chemokines and express ligands for diverse adhesion molecules leading to activation of various signaling pathways involved in survival and growth of HCL cells, i.e., MAPK pathways [61]. Herein, the bone marrow provides a protective microenvironment, therefore also being a common site of (minimal) residual disease. Strikingly, HCL cells can circumvent apoptosis induced either by interferon-α (IFN-α) or BRAF inhibitors when co-cultured with BMSCs [18,62]. Besides BMSCs, sinusoidal endothelial cells and stromal reticular cells play a critical role for the distinct bone marrow infiltration of HCL cells by secreting high levels of CXCL12, the ligand of CXCR4, a receptor highly expressed on HCL cells (Figure 1). Originally aimed at guiding the way of HSCs from the peripheral blood back to the protective bone marrow niche, HCL cells exploit this chemotaxis to benefit from these resourceful surroundings [63].

Furthermore, elevated serum levels of several other cytokines have been described in HCL patients and could be linked to disease activity in many cases, including interleukin-2 receptor, interleukin-1β and tumor necrosis factor alpha (TNF-α) [62,64,65]. Although the exact mechanism of action contributing to disease biology often remains a topic of ongoing research, autocrine TNF-α was demonstrated to induce expression of the anti-apoptotic proteins inhibitor of apoptosis (IAP) 1 and 2 through activation of the nuclear factor-κB pathway. Interestingly, IFN-α induced apoptosis of HCL cells can be referred to stimulating autocrine TNF-α production, but concomitantly downregulating IAP expression and, thus, sensitizing HCL cells to the pro-apoptotic effect of TNF-α [62].

The characteristic fibrosis seen in HCL composed of autonomous fibronectin secretion and reticulin fibers provided by surrounding fibroblasts illustrates another mechanism by which HCL cells benefit from their local microenvironment. Fibronectin synthesis is triggered by autocrine basic FGF production, which in turn responds to cell-to-matrix interaction with hyaluronan via CD44-binding [66,67]. Hence, HCL-associated fibrosis follows a distinct pattern of distribution with infiltration of hyaluronan-rich tissues such as the bone marrow and hepatic portal tracts, but sparing spleen and hepatic sinusoids. Reticulin fiber production by fibroblasts is stimulated by HCL cells’ secreted activated tumor growth factor beta1 (TGFß1, Figure 1) [68]. On the contrary, HCL associated splenomegaly is facilitated by extensive infiltration of the red pulp, where HCL cells form so-called pseudo-sinuses as a result of cell-to-cell interaction via the integrin subfamily β1 (i.e., integrin α4β1/VLA-4 and integrin α5β1) binding to local vitronectin as part of the extracellular matrix and vascular cell adhesion molecule (VCAM-1) on endothelial cells (Figure 1) [69]. Both interactions were shown to protect HCL cells from apoptosis when exposed to IFN-α or BRAF-inhibitors [18,62].

B-cell receptor (BCR) activation represents another mechanistic tool by which HCL cells receive important pro-survival signals and is increasingly recognized as a key pathogenic feature in several mature B-cell neoplasms. BCR downstream signaling via the Bruton’s tyrosine kinase (BTK) has been identified as a driving force in growth stimulation, cell survival as well as the homing properties of the malignant B-cells [70]. The relevance of BCR signaling in HCL is less well defined, although clear hints on a leading role in HCL biology date back to immunoglobulin analyses more than 10 years ago: Detection of mutated IGHV genes in nearly 90% of patients and restricted repertoire of IGHV gene usage, including IGHV3-21, IGHV3-30 and IGHV3-33, suggest some sort of selection pressure and indicate that BCR binding to similar, yet unknown, antigens may favor HCL transformation [12]. Interestingly, however, the minority of HCL cells with unmutated IGHV status seem more addicted to BCR stimulation as reflected by a more pronounced inhibition of relative phosphorylated protein levels when pretreated with the BTK inhibitor ibrutinib. In this regard, BTK inhibition has not only been shown to decrease MEK-ERK signaling strength but also to reduce secretion of chemokines such as CCL3 and CCL4 as well as downregulate CXCR4 signaling which is important for cell migration and homing (Figure 1) [63]. Strikingly, ibutinib, a first-in-class BTK inhibitor demonstrated encouraging efficacy in an anecdotal patient with relapsed refractory HCL variant that was recently confirmed in a phase-2 study including 37 patients with HCL or its variant [71,72].

An intriguing intersection of HCL cells’ inherent transformative triggers and crosstalk with the local tissue microenvironment may be HCL cells’ abnormal and diagnostic expression of the beta-2-integrin CD11c [11]. CD11c is part of a heterodimeric membrane receptor protein engaged in facilitating adherence of neutrophils and monocytes to activated endothelial cells [73]. As such, the gene encoding for CD11c is primarily active in cells of the myeloid lineage and, thus, aberrant and constitutive transcription in lymphocytic HCL cells has been suspected to be involved in HCL pathogenesis. Whereas a primary gene mutation could be widely excluded [23,24], CD11c promotor activity was demonstrated to highly depend on interaction with the activator protein-1 (AP-1) family of transcription factors. Interestingly, the AP-1 complex in HCL cells appears to contain JunD only, whereas in non-HCL cells AP-1 complexes are composed of a mixture of JunD, c-Jun, JunB and members of the Fos family [74]. Indeed, chronic JunD expression has been described in several other lymphomas, including adult T-cell leukemia and cutaneous T-cell lymphoma, both of which may co-occur with HCL in the same patient [75]. The cellular context regulating expression of the AP-1 family in HCL is not fully understood. However, fine-tuning of ERK and p38-JNK signaling upstream of AP-1 through by-standing feedback inhibitors at the transcriptional and posttranscriptional level has been demonstrated in various BRAF-mutated malignancies [18,32].

## 6. Immune Deficiency in Classic Hairy Cell Leukemia

The natural course of disease in HCL is frequently complicated by profound (opportunistic) infections [2]. Among the increasing hematopoietic insufficiency due to progressive bone marrow infiltration by the malignant clone, monocytopenia typically appears specifically pronounced. Although the underlying reasons remain to be fully elucidated, a context with the suppressive cytokine milieu provided by the HCL cells has been suggested, foremost attributable to TNF-α and TGF-β [76]. Thereby, monocytopenia may also at least in part account for an alteration of the T-cell compartment, possibly as a result of associated decreased antigen presentation and/or insufficient co-stimulation. In HCL, peripheral T-cell counts are typically not only reduced and exhibit a reversed CD4-CD8 ratio, but also show a decreased expression of costimulatory CD28 and restricted T cell repertoire, likely contributing to HCL-associated immune deficiency [77].

Besides monocytopenia, T-cell-related immune dysfunction may derive from interactions with the hairy cells *per se*. As is discussed above, yet unknown (auto)antigens driving constitutive B-cell receptor activation in HCL cells may also drive T-cell expansion and, thus, damp the T-cell receptor repertoire.

## 7. Conclusions

Over the last decade a great effort has been made to further elucidate the major biologic features underlying HCL. Although the activating BRAF V600E mutation could be identified as a key pathogenetic driver detectable in nearly all HCL patients [17], current data suggest that isolated BRAF mutations rather induce cell senescence and apoptosis than cell cycle progression and HCL transformation [26]. Hence, various whole-exome-sequencing studies aimed at tracking cooperating gene mutations facilitating the full phenotypic evolution of HCL. Besides recurrent inactivating mutations of the cell cycle inhibitor CDKN1Bb (p27) in 16% of patients postulated to help BRAF V600E-mutated malignancies escape cell senescence [24], other identified inactivating gene mutations suggest an even more heterogeneous pattern of genetic alterations, each being detectable only in a small subgroup of HCL patients [23]. As such, cooperating biologic features in a phenotypically homogeneous and genetically stable disease may rather be localized at the epigenetic level. Genome-wide expression profiling studies consistently demonstrated an elevated transcriptional output of the MEK/ERK pathway through disabled feedback inhibition in HCL and other BRAF-mutated malignancies, including dysregulation of the DUSP family (i.e., DUSP6) [18,32]. DUSP subtypes were also shown to mediate crosstalk with other dysregulated MAPK pathways implicated in HCL pathogenesis such as JNK and p38. JNK and p38 pathway activation triggers are primarily derived from extracellular stimuli through cytokines and G-protein coupled receptors [78]. As reflected by their distinct pattern of tissue infiltration and increased cell surface area, HCL cells highly depend on various extracellular stimuli provided by their local tissue microenvironment and may even escape treatment-induced cell death when co-cultured with stromal compounds [18,62]. In the end, the biologic features underlying HCL may involve both genetic and epigenetic alterations with a strong dependence on extracellular stimuli. Besides the nearly ubiquitously detectable BRAF V600E mutation, the impact of epigenetic alterations and extracellular triggers may vary among individual patients—an issue that remains to be further addressed in functional studies. The introduction of DNA and RNA single-cell sequencing studies of HCL cells and their local microenvironment may provide valuable insights to define the malignant cells’ clonal composition and better dissect the pathways involved in the crosstalk of HCL and their resource-rich surroundings. Ultimately, these studies may also help to identify the additional biologic features driving HCL besides the recurrent BRAF V600E mutation.

## Figures and Tables

**Figure 1 ijms-22-07780-f001:**
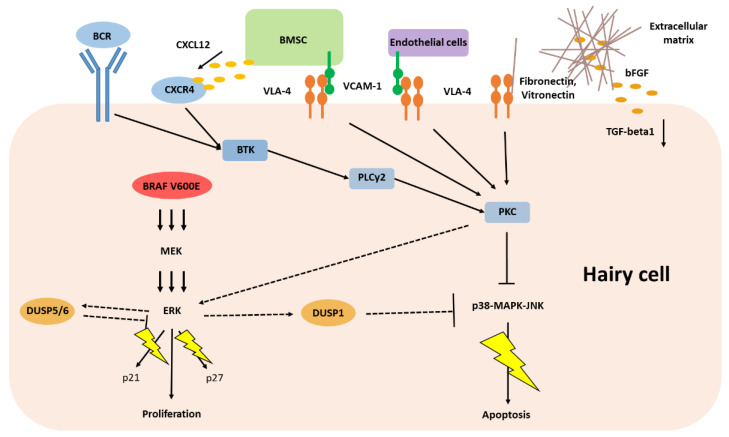
Current understanding of classic hairy cell leukemia biology. Line arrows with thunder illustrate evasion from pro-apoptotic signaling pathways; Dash arrows show proposed signaling outputs of constitutive ERK-signaling derived from preclinical data in BRAF-mutant melanoma cell lines; BRAF V600E, driver mutation in HCL; BCR, B-cell receptor; bFGF, basic fibroblast growth factor; BMSC, mesenchymal bone marrow stromal cells; BTK, Bruton’s tyrosine kinase; CXCL12, C-X-C motif chemokine 12; CXCR4, C-X-C chemokine receptor 4; DUSP, dual specificity phosphatase; ERK, extracellular-signal regulated kinase; JNK, c-Jun N-terminal kinase; MAPK, mitogen-activated protein kinase; MEK, mitogen-activated protein kinase kinase; PKC, protein kinase C; PLCy, phospholipase C gamma; TGF, tumor growth factor; VCAM, vascular cell adhesion molecule; VLA, very late antigen, adhesion molecules.

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
