# Peer review of "The Biology of Classic Hairy Cell Leukemia"

_ijms, 2021, doi:10.3390/ijms22157780_

Round 1

Reviewer 1 Report

The article entitled " The biology of classic hairy cell leukemia (HCL) is interesting and useful for the medical community, which is  interested in HCL and HCL like disorders.

For improving the manuscript, a few comments/suggestions

1) you must shorten the manuscript, which is to much difficult to read and adapted to specialists, +++

2) from the biologic data, introduce the interest of the data from a clinical point of view and the drugs we could use if the biologic effect is effectively relevant, +++

3) the chapter 3 is important but in the present form the role of DUSPs are confusing and the messages, complex and not sure.

4) the chapter 4 is not sufficiently pedagogic and too general. You could reduce it.

5) For the microenvironnment could you specify the role of the monocytes? 

Author Response

Detailed point-by-point reply:

Response to Reviewer 1

1) you must shorten the manuscript, which is to much difficult to read and adapted to specialists, +++

à We apologize for being too detailed and now shortened the original manuscript by 500 words. According to your recommendation (see point 4), we mainly reduced the length of chapter 4.      Novel additions recommended by the second reviewer are excluded.

2) from the biologic data, introduce the interest of the data from a clinical point of view and the drugs we could use if the biologic effect is effectively relevant, +++

à Thank you for this constructive and important comment. We now included clinical trial data for BRAF- and MEK- (see lines 87-92), DUSP- (see lines 184-186) and BTK-inhibitors (see lines 320-322) in HCL.

3) the chapter 3 is important but in the present form the role of DUSPs are confusing and the messages, complex and not sure.

à We apologize that this chapter was not clearly written, we therefore content-wise revised chapter 3. In particular, we rephrased lines 174 to 180 to review this important data more clearly.

4) the chapter 4 is not sufficiently pedagogic and too general. You could reduce it.

à Please see  our response to point 1.

5) For the microenvironnment could you specify the role of the monocytes? 

à As suggested by the assigned editor, we added another section about immune deficiency in HCL. In lines 334-358 of the revised manuscript, we now discuss potential multifactorial mechanisms acting in concert to inhibit the immune competence of HCL patients. This also includes the role of HCL-related monocytopenia and T-cell dysfunction.

Reviewer 2 Report

The manuscript “The biology of classic hairy cell leukemia” by Bohn et al, is a narrative review article submitted for the IJMS Special Issue “Immunological Investigations in Hematology”. In the manuscript authors summarize “the current understanding of key pathogenic mechanisms implicated in hairy cell leukemia and discusses major hurdles to overcome in the context of other BRAF-mutated malignancies”.

The work builds upon the extensive experience on oncology maturated by Prof. Wolf and coworkers.

The current review is well presented and organized.

I suggest minor points authors might address to further improve the potential impact of the manuscript.

  • Some studies have described chromosomal aberrations in classical HCL and in its variant, but their significance is unclear. Could authors address this point?
  • Assessment of TP53 status in classic HCL has been suggested. Could authors provide recent update on this issue?
  • Albeit the cohort of patients was limited, epigenetic regulation genes mutation have been described in a portion of classic HCL patients (Maitre et al, 2018 doi: 10.18632/oncotarget.25601).
  • Alterations in levels of different cytokines have been described in the sera from patients with HCL. Influence of paracrine signaling influencing the biology of HCL should be addressed in more details in addition to local microenvironmental cues.
  • I noticed that the most recent cited reference has been published on March 2019. I wonder whether authors might be missing more recent papers to reflect the current the state of the art of hairy cell leukemia basic and translational research.

Author Response

Detailed point-by-point reply:

Response to Reviewer 2

I suggest minor points authors might address to further improve the potential impact of the manuscript.

  • Some studies have described chromosomal aberrations in classical HCL and in its variant, but their significance is unclear. Could authors address this point?

à Thank you for this constructive comment. To maintain the structure of the manuscript, we reviewed the recurrent copy number loss of chromosome 7p reported in one recent sequencing study and its implication in lines 97-104. Deletion 13q, seen in a minor fraction of patients and its implications has been added in lines 231-234.

  • Assessment of TP53 status in classic HCL has been suggested. Could authors provide recent update on this issue?

à Thank you for this meaningful addition.  We now reviewed TP53 incidence in HCL and assessment recommendations in lines 120-123.

  • Albeit the cohort of patients was limited, epigenetic regulation genes mutations have been described in a portion of classic HCL patients (Maitre et al, 2018 doi: 10.18632/oncotarget.25601).

à  Thank you very much for this important and helpful note! We now added this data into the revised review version (see lanes 115-119).

  • Alterations in levels of different cytokines have been described in the sera from patients with HCL. Influence of paracrine signaling influencing the biology of HCL should be addressed in more details in addition to local microenvironmental cues.

à Thank you very much for shedding light onto these important studies. Besides signaling via CXCl12, hyaluronan, FGF, TGF-beta, vitronectin and fibronectin, we now also included reviewing the role of interleukin 1 and 2 as well as TNF-alpha (see lines 278-288).

  • I noticed that the most recent cited reference has been published on March 2019. I wonder whether authors might be missing more recent papers to reflect the current the state of the art of hairy cell leukemia basic and translational research.

à Thank you very much for underlining the paucity of significant novel findings on HCL biology in very recent years. Searching the data bases such as pubmed, recent publications related to classic HCL mainly focused on investigating new drugs such as BRAF-inhibitors, MEK-inhibitors and BTK-inhibitors. As recommend by the second reviewer, these novel clinical trial data has now also been added to our revised manuscript. However, using new technologies such as single-cell RNA sequencing, investigators dedicated to HCL are eagerly looking for new biologic insights that may ultimately help to draw the last secrets from HCL biology.

Round 2

Reviewer 1 Report

Thanks for your modifications

The manuscript is clear and pedagogic.